# Effects of Pressure Control Device (SensAwake™) on Obstructive Sleep Apnea (OSA) Patients Who Remove the Mask for Unknown Reasons during Automatic Continuous Positive Airway Pressure (Auto-CPAP) Therapy: A Prospective Randomized Crossover Trial

**DOI:** 10.3390/medicina57090915

**Published:** 2021-08-31

**Authors:** Yen-Lung Chen, Li-Pang Chuang, Shih-Wei Lin, Hung-Yu Huang, Geng-Hao Liu, Hung-Fu Hsu, Ning-Hung Chen

**Affiliations:** 1Sleep Center, Chang Gung Memorial Hospital, Taoyuan 33378, Taiwan; 8905029@cgmh.org.tw (Y.-L.C.); ec108146@cgmh.org.tw (S.-W.L.); b9202071@cgmh.org.tw (H.-Y.H.); 8905033@cgmh.org.tw (G.-H.L.); nhchen@cgmh.org.tw (N.-H.C.); 2Center for Traditional Chinese Medicine, Chang Gung Memorial Hospital, Taoyuan 33378, Taiwan; 3Graduate Institute of Clinical Medical Sciences, College of Medicine, Chang Gung University, Taoyuan 33302, Taiwan; 4Department of Pulmonary and Critical Care Medicine, Chang Gung Memorial Hospital, Taoyuan 33305, Taiwan; 5Department of Industrial Design, Chang Gung University, Taoyuan 33302, Taiwan; hidrtommy@mail.cgu.edu.tw; 6Department of Respiratory Therapy, Chang Gung University, Taoyuan 33302, Taiwan

**Keywords:** continuous positive airway pressure, nasal obstruction, disorders of excessive somnolence, obstructive sleep apnea

## Abstract

*Background**and Objectives*: Obstructive sleep apnea (OSA) patients may remove their mask unconsciously during automatic continuous positive airway pressure (Auto-CPAP) therapy and therefore cannot receive good treatment. The discomfort from the airflow of Auto-CPAP may be one reason for interrupted sleep. Sens Awake (SA) can detect the arousal and lower the pressure to prevent patients from fully awakening from sleep. *Materials and Methods*: To evaluate the effect of SA, we designed a prospective, randomized, crossover trial comparing Auto-CPAP with and without SA on Epworth Sleepiness Scale (ESS), Pittsburgh Sleep Quality Index (PSQI), Nasal Obstruction Symptom Evaluation (NOSE) Scale and recorded data from the auto-CPAP machine. *Results*: In the 25 patients who completed the study, the gender, age, body mass index, neck circumference, polysomnography data, and previous CPAP use were not significantly different between the two arms. The average and 90th percentile pressures were significantly lower during SA on (SA on vs. off: 6.9 ± 2.7 vs. 7.3 ± 2.6 [*p* = 0.032] and 8.6 ± 3.0 vs. 9.2 ± 2.9 [*p* = 0.002], respectively). The time used, days used, compliance, average and 90th percentile leaks, and the residual Apnea-Hypopnea Index (AHI) were not significantly changed between the SA on-and-off. Based on the subjective evaluation, PSQI, ESS, and NOSE were not significantly different between the SA on-and-off; however, based on additional analyses which were compared with baseline data, the ESS was significantly lower when the SA was on (SA on vs. baseline: 11.1 ± 6.1 vs. 13.2 ± 6.0 [*p* = 0.023]). *Conclusions*: CPAP therapy with or without two weeks of the SA had a similar effect on CPAP use, sleep quality, daytime sleepiness, and nasal obstruction. The SA may have a tendency to improve daytime sleepiness, but needs further study with a longer duration of treatment.

## 1. Introduction

Obstructive sleep apnea syndrome (OSA) is prevalent and has severe consequences without adequate treatment [1]. Patients with OSA may experience daytime sleepiness, poor concentration and memory, and the long-term effects of OSA are associated with cognitive impairment and increased all-cause and cardiovascular disease-specific mortality [2]. Continuous positive airway pressure (CPAP) ventilator therapy is the first-line treatment for OSA patients and is effective in improving OSA symptoms [3]. Furthermore, CPAP is effective in controlling blood pressure in hypertensive patients [4], reducing the cardiovascular disease-related morbidity and mortality rates, and improving long-term outcomes [1].

The continuity of CPAP use has a dose-dependent effect on the consequences of OSA [5]; however, some OSA patients may remove the mask themselves while sleeping, which leads to poor compliance and an insufficient CPAP therapy effect [6]. The well-known reason that patients unconsciously remove the mask is increased nasal resistance, which results in a subjective nasal congestion with CPAP intolerance, and, consequently, patients take off their mask [7]. Other reasons for unconscious mask removal may reflect inappropriate pressure settings, flow leakage, or discomfort from the mask, but supporting evidence is lacking. Repetitive removal of the mask can lead to poor sleep quality, insufficient therapy effect, poor compliance, and even discontinuation of CPAP therapy.

To improve the above conditions, advanced PAP features, including auto-titrating CPAP [8,9], bi-level PAP [10,11,12], and expiratory pressure relief [13,14,15,16], were developed to decrease discomfort from high-pressure airflow that may reduce patient compliance with recommended treatment. High pressure may lead to patient discomfort, and unconscious removal of the mask when awakening from sleep. We hypothesize that relieving the pressure during arousal can reduce the uncomfortable sensation and facilitate prompt return to sleep without removal of the mask.

Ayappa et al. reported that analysis of the respiratory pattern may be a useful indicator of detection of wake or arousal [17]. The technology SensAwake™ (SA) reducing the pressure when detect the patient is awake is thought to prevent awakening induced by high pressure [18], which is used in the newly designed auto-CPAP machine by Fisher & Parkel Healthcare, Ltd., Auckland, New Zealand. In a randomized crossover trial by Dungan and colleagues [19], single night of SA has no significant improvement in wake after sleep onset (WASO) and other sleep architecture measures, and longer-term studies are needed. Pepin and colleagues [20] thought SA may improve objective sleep quality in OSA combined with insomnia. To confirm the effect of SA on the continuity of CPAP and improving the consequences of OSA, we designed the current study to collect patients who unconsciously remove their masks with poor continuity of CPAP, and randomly used SA on-and-off.

## 2. Materials and Methods

We conducted a prospective, randomized, cross-over study to evaluate the effect of pressure relief in CPAP therapy. Participants who were > 18 years of age with moderate-to-severe OSA (Apnea Hypopnea Index [AHI] > 15/h) and received CPAP treatment for the first time for at least one month, but had poor continuity without a known etiology, were invited to participate in this study. Poor continuity was defined as removal of the mask without a known reason. Participants who removed the mask for a known reason, such as mask leakage, high pressure, nasal congestion, skin problem, or humidifier problem were excluded. In addition, participants who had unstable cardiovascular disease (untreated or resistant hypertension was acceptable), inability to tolerate CPAP due to nasal obstruction or claustrophobia as determined by the study investigator, or any known factor or disease that might interfere with treatment compliance, study conduct or interpretation of the results, such as psychiatric disease, a history of non-compliance to medical regimens or unwillingness to comply with study requirements, were excluded from this study.

Participants were randomized into two arms. Specifically, participants used CPAP with or without activation of SA for 2 weeks, then crossed over to the other arm for an additional two weeks. All participants were asked to fulfill three questionnaires. The Epworth Sleepiness Scale (ESS) [21] contains 8 questions and the scored range from 0 to 24. In general, a score greater than ten was defined as excessive daytime sleepiness. The Pittsburgh Sleep Quality Index (PSQI) [22] contains 19 self-rated questions and 5 questions rated by bed partner or roommate to evaluate the sleeping quality. A total score more than 5 is associated with poor sleep quality. The Nasal Obstruction Symptom Evaluation (NOSE) Scale [23] contains 11 items that scored range from 0 to 40 (one item is general healthy related and did not score) to evaluate nasal obstructive condition. The higher the score means the poorer nasal obstructive condition. Objective data from the auto-CPAP machine, including time used, percentage of day used, compliance, average and 90th percentile pressures, average and 90th percentile leaks and residual AHI, were recorded.

Data were collected at baseline, and the end of the 2nd and 4th weeks. Data analyses was performed using a statistical software package (IBM Corp. Released 2011. IBM SPSS Statistics for Windows, Version 20.0. Armonk, NY: IBM Corp). The baseline data were presented as mean ± standard deviation (SD) and the comparisons were made using the Mann-Whitney U test. The data of SA ON and OFF were presented as median and interquartile range (IQR) and the comparisons were made using the Wilcoxon sign-rank test. The significance level was 0.05.

The sample size was calculated based on score of ESS. We expected that ESS could be improved from moderate (ESS = 13) to higher normal level (ESS = 10). We used G*power (Version 3.1.9.7 for windows) with a setting of power = 0.8, two-tailed alpha = 0.05 and correlation between groups = 0.5, and got the total sample size of 24 subjects (12 pairs in crossover study). We estimated a 30% sample size missing rate, therefore, it needed at least 35 subjects enrolled in this study.

Written informed consent was obtained from all participants. This study was approved by the Chang Gung Medical Foundation Institutional Review Board (approval number 103-7083B) and registered at http://clinicaltrials.gov (accessed on 10 July 2021). (Identifier: NCT03294629).

## 3. Results

We screened 415 OSA patients and who were recommended to receive CPAP therapy; 249 patients agreed to CPAP therapy, while the remaining patients chose to research other treatment options. One hundred twenty-one patients discontinued CPAP therapy after 1 month, and the other 127 patients continued to receive CPAP therapy. We inquired about the status of CPAP therapy in recent days, and selected patients who removed their masks during CPAP therapy for unknown reasons, including an inability to adapt to CPAP therapy, nasal congestion, conscious removal of the mask, and awakening during therapy. Thirty-five patients who removed their masks without a known etiology were invited to participate in this study. Of the 35 participants, 16 were in arm 1 and began with the SA on, and the other 19 participants in arm 2 began with the SA off. Ten participants were unwilling to continue or receive CPAP therapy and withdrew from the study; 25 participants completed the study (Figure 1). The general baseline data are shown in Table 1. The gender, age, body mass index, neck circumference, polysomnography data, and previous CPAP use were not significantly different between the two arms.

Table 2 presents the differences between SA on-and-off. The average and 90th percentile pressures were significant lower during SA on (SA on vs. off: 6.9 ± 2.7 vs. 7.3 ± 2.6 [*p* = 0.032] and 8.6 ± 3.0 vs. 9.2 ± 2.9 [*p* = 0.002], respectively). The time used, days used, compliance, average and 90th percentile leaks, and the residual AHI were not significantly changed between the SA on-and-off. Based on the subjective evaluation, PSQI, ESS, and NOSE were not significantly different between the SA on-and-off; however, based on additional analyses which were compared with baseline data, the ESS was significantly lower when the SA was on (SA on vs. baseline: 11.1 ± 6.1 vs. 13.2 ± 6.0 [*p* = 0.023]; Figure 2).

## 4. Discussion

In the current study, activation of the SA reduced the pressure of CPAP, and the residual AHI simultaneously increased significantly, and thus the function of detecting wakefulness and relieving the pressure occurred during CPAP therapy; however, the SA did not improve the time used, days used and compliance with CPAP use. Furthermore, the sleep quality, daytime sleepiness and nasal obstruction did not improve with the SA. Patients remove their masks for multiple causes, and not just due to high-flow pressure. The patients enrolled in this study all had experience removing their masks while asleep, but they did not know why, although the pressure of flow may have been one cause. Hollandt and colleagues [24] reported that patients with nasal obstruction and rhinorrhea after using nCPAP are unable to breathe through their nose and unconsciously remove their masks. Other reasons to remove the mask may include the noise emanating from the CPAP device, discomfort from the mask, and/or flow leakage, but these causes have not been verified. Patients may not actually know what occurs while sleeping, and awake to find their mask removed. Patients may remove the mask when they awaken, but spontaneously change their position while sleeping; the SA is less effective in such situations. Due to above reason, although SA could decrease the influence from high-flow pressure, other causes that lead to mask removal still needed to be discovered and improved.

In current study, we used PSQI, ESS and NOSE to assess subjects’ sleeping quality, daytime excessive sleepiness and nasal congestion, respectively. Other tools to evaluate OSA including Berlin Questionnaire [25], Snore Outcome Survey (SOS) [26] and Modified Mallampati score (MMP) [27]. Berlin Questionnaire and SOS could evaluate improvements of subjective snoring condition. MMP together with upper airway volume and the Berlin Questionnaire might be reliable indicators to assess the risk of snoring [28]. The above mentioned tools could be used to assess the effects of SA in further study.

In a previous study by Dungan and colleagues [19], no measure of sleep architecture was significantly improved by the SA mode during a single night exposure with evaluation by polysomnography. Even activation of the SA can detect wakefulness, but the time of awakening after sleep onset (WASO) did not significantly improve with the SA. No outcome measure is significantly affected by the SA, with the exception of technician satisfaction with the therapy, which is significantly lower under SA conditions [18]. In the current study, the ESS was lower in the SA on than the SA off mode, but there was no significant difference. Previous study by Bogan and colleagues [29] also found no significant difference in ESS compared with SA ON and OFF. Based on an additional analysis, the SA significantly improved daytime sleepiness compared with baseline data. Daytime sleepiness is a common problems in OSA patients, well CPAP adherence and compliance can eliminate daytime sleepiness by providing sufficient airflow during sleeping. The SA may decrease the number of arousals, sleep latency when back to sleep after awakening, percentage of shallow sleep and needed be confirmed by polysomnography. In our study, the SA may have potential to improve daytime sleepiness, and two-weeks of intervention might not be sufficient and the sample size was small. Therefore, a corollary study enrolling additional participants, with polysomnography evaluation during SA ON and OFF, and a longer treatment period may confirm the benefits of the SA.

Before enrolling in the current study, the residual AHI of participants was <5 during CPAP treatment and reached the therapeutic goal of CPAP; however, the residual AHI increased during CPAP treatment with the SA on. This result was mainly due to the relief of pressure by the SA when detecting patient wakefulness, and can lead to an increase in AHI at the same time. If we can use a lower pressure to maintain the treatment effect and offer less discomfort to the patient, the SA may also benefit the patient. Further study can evaluate the long-term treatment effects on blood pressure changes, severity, morbidity and mortality rates of cardiovascular disease with lower CPAP pressure and the SA ON.

A limitation of this study was the lack of a washout period following cross-over to the other treatment arm. The participants in the current study all received CPAP therapy for at least one month, and we could not ask the participants to discontinue CPAP therapy for ethical reasons, so the results may be affected by previous therapy (i.e., whether or not the SA is on or off).

## 5. Conclusions

CPAP therapy with or without two weeks of the SA had a similar effect on CPAP use, sleep quality, daytime sleepiness and nasal obstruction. The SA may have a tendency to improve daytime sleepiness, but needs further study with a longer duration of treatment.

## Figures and Tables

**Figure 1 medicina-57-00915-f001:**
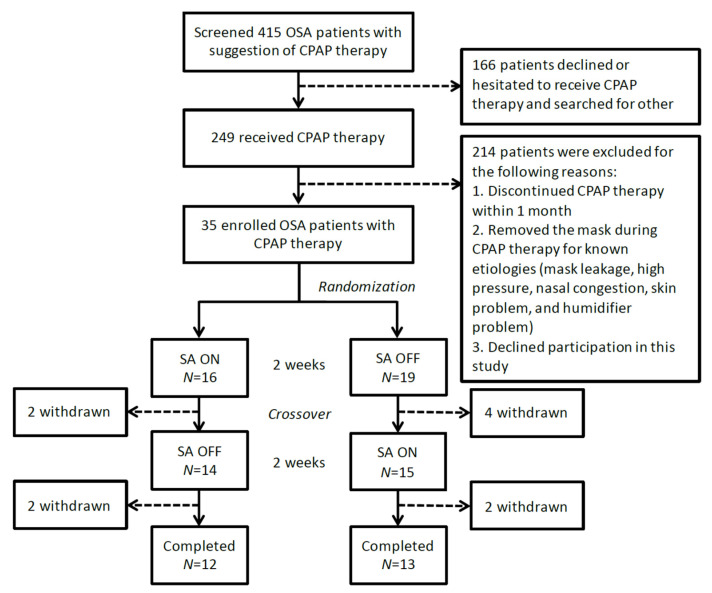
Patient flow diagram. Thirty-five patients were enrolled in the study and were randomized. Ten withdrew and 25 patients completed the crossover study. OSA = obstructive sleep apnea, CPAP = continuous positive airway pressure, SA = SensAwake^TM^.

**Figure 2 medicina-57-00915-f002:**
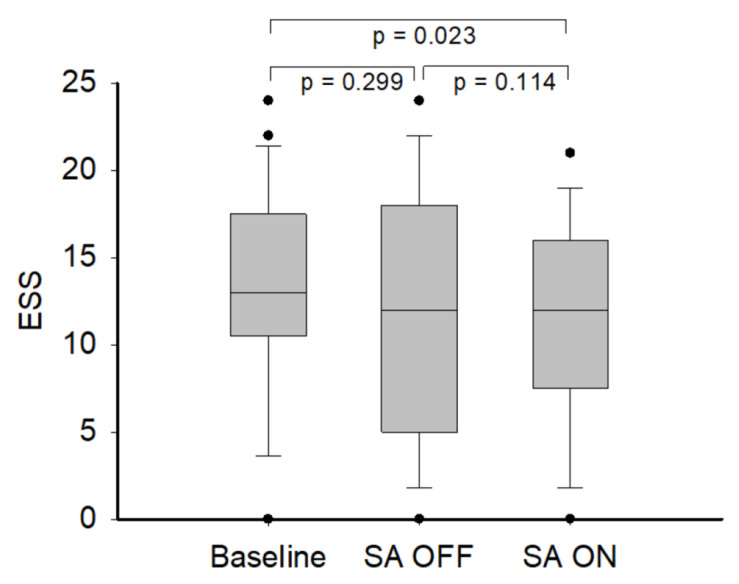
Comparison of baseline, SA OFF, and SA ON in ESS (*n* = 25). The horizontal line is the median, box is the IQR and whiskers are the 95% range. *p* values were from Wilcoxon sign-rank test. The ESS when the SA was on were significantly lower than baseline. ESS = Epworth sleepiness scale, SA = SensAwake^TM^, IQR = interquartile range.

**Table 1 medicina-57-00915-t001:** Baseline data between the two arms.

	Arm 1 (*n* = 12) SA ON First	Arm 2 (*n* = 13) SA OFF First	*p* Value
General data			
Gender: Female (%)	0 (0.0)	3 (23.1)	0.220
Age	47.6 ± 10.9	41.5 ± 12.0	0.207
BMI	31.8 ± 4.3	32.9 ± 6.9	0.642
NC	42.3 ± 2.1	41.0 ± 2.2	0.167
Polysomnography			
AHI	55.3 ± 30.8	54.0 ± 35.3	0.923
Sleep efficiency (%)	82.0 ± 8.9	73.0 ± 21.3	0.183
Total sleep time (min)	301.0 ± 34.2	276.2 ± 79.2	0.347
Deep sleep—Stage 3 (%)	6.1 ± 10.4	4.4 ± 5.6	0.624
CPAP device recorded data			
Average used time (min)	187.4 ± 104.6	227.9 ± 104.1	0.342
Compliance (%)	31.9 ± 31.9	46.2 ± 30.6	0.266
Residual AHI	2.9 ± 1.3	4.4 ± 2.8	0.101
Self-administered questionnaire			
PSQI	7.4 ± 2.3	10.0 ± 3.8	0.053
ESS	13.3 ± 5.5	13.1 ± 6.7	0.918
NOSE	13.5 ± 6.8	15.6 ± 8.4	0.498

The general data, polysomnography parameters, previous CPAP use, and three questionnaires were not significantly different between the two arms. The comparisons were made using the Mann-Whitney U test. Data was presented as mean ± SD and a *p*-value less than 0.05 was statistically significant. SA = SensAwake^TM^, BMI = body mass index, NC = neck circumference, AHI = apnea-hypopnea index, CPAP = continuous positive airway pressure, PSQI = Pittsburgh sleep quality index, ESS = Epworth sleepiness scale, NOSE = nasal obstruction symptom evaluation scale.

**Table 2 medicina-57-00915-t002:** Comparison of SensAwake^TM^ ON and OFF during CPAP therapy. (*n* = 25).

	SA ON	SA OFF	*p* Value
CPAP device recorded data			
Average SA detections (/h)	1.9 (1.5–2.6)	N/A	-
Pressure	6.0 (5.0–8.5)	6.5 (5.0–9.0)	0.063
90% pressure	8.0 (6.3–10.0)	8.5 (6.8–10.5)	0.004 *
Leak	31.0 (27.5–54.5)	43.0 (26.0–54.5)	0.367
90% leak	57.0 (35.5–79.0)	49.0 (34.0–80.0)	0.626
Residual AHI	3.9 (3.3–6.3)	3.8 (2.4–6.5)	0.017 *
Average use (min)	236.0 (111.0–293.0)	193.0 (124.0–285.5)	0.904
Days used (%)	93.0 (74.0–100.0)	92.0 (79.5–100.0)	0.681
Compliance (%)	36.0 (4.0–81.0)	31.0 (9.5–69.0)	0.972
Self-administered questionnaire			
PSQI	8.0 (6.0–10.8)	8.0 (5.5–11.0)	0.924
ESS	12.0 (7.5–16.0)	12.0 (5.0–18.0)	0.121
NOSE	12.0 (7.5–23.5)	12.0 (8.5–18.5)	0.245

The average and 90th percentile pressures and residual AHI were significant lower during SA ON. Other parameters were not significantly changed between the SA ON-and-OFF. The comparisons were made using the Wilcoxon sign-rank test. Data was presented as Median (IQR) and a *p*-value less than 0.05 was statistically significant. * *p* < 0.05, CPAP = continuous positive airway pressure, SA = SensAwake^TM^, AHI = apnea-hypopnea index, PSQI = Pittsburgh sleep quality index, ESS = Epworth sleepiness scale, NOSE = nasal obstruction symptom evaluation scale, IQR = interquartile range.

## Data Availability

Data sharing is not applicable to this article.

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
