# Peer review of "Effects of Pressure Control Device (SensAwake™) on Obstructive Sleep Apnea (OSA) Patients Who Remove the Mask for Unknown Reasons during Automatic Continuous Positive Airway Pressure (Auto-CPAP) Therapy: A Prospective Randomized Crossover Trial"

_medicina, 2021, doi:10.3390/medicina57090915_

Round 1

Reviewer 1 Report

This is an interesting study concerning Pressure Control Device on Obstructive Sleep Apnea patients reluctant of auto-CPAP therapy. Topic is important, as sleep apnea is a growing health problem, with potentially deadly complications, yet continuous positive airway pressure therapy devices are definitely solutions, which will remain with us for a longer period of time. However, some criticism should be raised prior :

Introduction

L42-43 - First sentence requires relevant citation(s) - consider moving [1] here from L45

Materials and methods

L76 - Was the study registered and was number obtained? Please provide these data.

L90-94 - these questionnaires should be briefly described in Introduction section, also authors should explain why exactly these ones were chosen for the study as there are few more available; also Berlin questionnaire as it is becoming valuable screening tool, together with Mallampati score thus please incorporate and cite https://www.mdpi.com/2076-3417/11/9/3764

L100 - software manufacturers' data is missing, please provide

Table 1, 2 and Figure 1 results require short description within text

Reviewer 2 Report

This manuscript is a RCT on the effects of pressure control device (SensAwake) on OSA patients who remove theirs masks during auto-CPAP therapy.

It is an interesting work, a well written paper. Some minor improvements:

However, I suggest some minor changes:

Title:

As may guidelines state, I suggest that the title should include that this a “prospective, randomized, crossover trial”.

Abstract:

SA is used without explaining its meaning.

I suggest to do so in line 21: … “SensAwake  (SA)can detect the arousal and lower the pressure and thought…

Introduction

The authors introduce the use of SensAwake as a pressure control device (line 68). However, they do not quote any references. They should do at this point.

Killick R, Marshall NS. The Impact of Device Modifications and Pressure Delivery on Adherence. Sleep Med Clin. 2021 Mar;16(1):75-84. doi: 10.1016/j.jsmc.2020.10.008.

Ref 22 SHOULD BE QUOTED EARLIER: Dungan GC 2nd, Marshall NS, Hoyos CM, Yee BJ, Grunstein RR. A randomized crossover trial of the effect of a novel method of pressure control (SensAwake) in automatic continuous positive airway pressure therapy to treat sleep disordered breathing. J Clin Sleep Med. 2011 Jun 15;7(3):261-7. doi: 10.5664/JCSM.1066.

Mat & Methods

Sample size calculation is missing. Why only 35 participants were enough? Please provide it.

Results

In lines 105-111 and Figure 1, the authors explain why they only chose 35 patients to be randomized.

“We screened 415 OSA patients and who were recommended to receive CPAP therapy; 249 patients agreed to CPAP therapy, while the remaining patients chose to research other treatment options. One hundred twenty-one patients discontinued CPAP therapy after 1 month, and the other 127 patients continued to receive CPAP therapy. We inquired about the status of CPAP therapy in recent days, and selected patients who removed their masks during CPAP therapy for unknown reasons, including an inability to adapt to CPAP therapy, nasal congestion, conscious removal of the mask, and awakening during therapy. Thirty-five patients who removed their masks without a known etiology were invited to participate in this stud”

However, out of 415 patients, having only 127 on CPAP seems to be a low percentage. Please explain such low adherence rate.

Discussion

The main results is that activation of SensAwake reduced the pressure of CPAP, but the residual AHI simultaneously increased significantly.

As they remark in lines 143-145, the function of detecting wakefulness and relieving the pressure occurred during CPAP therapy; however, the SA did not improve the time used, days used, and compliance with CPAP use.

Please, further explain if this is actually clinically relevant.

Similarly, in lines 162-166: the ESS was lower in the SA on than the SA off mode, but there was no significant difference. Based on an additional analysis, the SA significantly improved daytime sleepiness compared with baseline data (Figure 2). Thus, the SA may have potential to improve fragmented sleep, but this has not been clearly demonstrated.

Please, again, further explain if this is actually clinically relevant.

I suggest them to include also the following VERY SIMILAR papers:

Bogan RK, Wells C. A Randomized Crossover Trial of a Pressure Relief Technology (SensAwake™) in Continuous Positive Airway Pressure to Treat Obstructive Sleep Apnea. Sleep Disord. 2017;2017:3978073. doi:10.1155/2017/3978073

Pepin JL, Gagnadoux F, Foote A, Vicars R, Ogra B, Viot-Blanc V, Benmerad M, D'Ortho MP, Tamisier R. Combination of obstructive sleep apnoea and insomnia treated by continuous positive airway pressure with the SensAwake pressure relief technology to assist sleep: a randomised cross-over trial protocol. BMJ Open. 2017;7(10):e015836. Published 2017 Oct 27. doi:10.1136/bmjopen-2017-015836

Reviewer 3 Report

The study by Chen et al. that explores Effects of Pressure Control Device (SensAwake™) on Obstructive Sleep Apnea (OSA) Patients Who Remove the Mask for Unknown Reasons during Automatic Continuous Positive Airway Pressure (auto-CPAP) Therapy is interesting, however it has several major issues:

The number of subjects is very limited, so it is difficult to have strong conclusions due to low power of the study.

In the methods section the authors should specify the level of significance used. Additionally, the authors should state with which test was normality of distribution assessed. If Wilcoxon signed-rank test was used (a non-parametric test) than the data cannot be presented with mean and SD, but with median and IQR.

Statistical test used should be noted in footnotes of tables and figures. Please specify the N for each of the groups in figure 2 as it is not clear what does baseline represent.

Why did the authors used 2 wk period, it seems too short to spot the differences in sleep quality, especially considering that there was no wash out period?

Why didn’t the authors discuss the observed difference in ESS between baseline and SA ON. It is doubtful that is clinically relevant, yet it is significant.

Round 2

Reviewer 3 Report

I do not have any further comments.

This manuscript is a resubmission of an earlier submission. The following is a list of the peer review reports and author responses from that submission.